# Epidemiological Characterization of Isolates of *Salmonella enterica* and Shiga Toxin-Producing *Escherichia coli* from Backyard Production System Animals in the Valparaíso and Metropolitana Regions

**DOI:** 10.3390/ani13152444

**Published:** 2023-07-28

**Authors:** Constanza Urzúa-Encina, Bastián Fernández-Sanhueza, Erika Pavez-Muñoz, Galia Ramírez-Toloza, Mariela Lujan-Tomazic, Anabel Elisa Rodríguez, Raúl Alegría-Morán

**Affiliations:** 1Departamento de Medicina Preventiva Animal, Facultad de Ciencias Veterinarias y Pecuarias, Universidad de Chile, Santa Rosa 11735, La Pintana, Santiago 8820808, Chile; constanza.urzua23@gmail.com (C.U.-E.); bastian.fernandez.s@ug.uchile.cl (B.F.-S.); isabelpm95@gmail.com (E.P.-M.); galiaram@uchile.cl (G.R.-T.); 2Laboratorio Centralizado de Investigación Veterinaria, Facultad de Ciencias Veterinarias y Pecuarias, Universidad de Chile, Santa Rosa 11735, La Pintana, Santiago 8820808, Chile; 3Escuela de Medicina Veterinaria, Sede Santiago, Facultad de Recursos Naturales y Medicina Veterinaria, Universidad Santo Tomás, Ejercito Libertador 146, Santiago 8370003, Chile; 4Instituto de Patobiología Veterinaria, Instituto Nacional de Tecnologías Agropecuarias, Consejo Nacional de Investigaciones Científicas y Técnicas, Av. de los Reseros y Nicolás Repetto s/n, Hurlingham, Buenos Aires 1686, Argentina; tomazic.mariela@inta.gob.ar (M.L.-T.); rodriguez.anabel@inta.gob.ar (A.E.R.); 5Facultad de Farmacia y Bioquímica, Universidad de Buenos Aires, Av. Junín 954, Buenos Aires C1113 AAD, Argentina

**Keywords:** *Salmonella enterica*, STEC, risk factors, antimicrobial resistance, positivity rate, backyard production system, One Health

## Abstract

**Simple Summary:**

This study aimed to investigate the prevalence and risk factors for *Salmonella enterica* and Shiga toxin-producing *Escherichia coli* (STEC) in backyard production systems (BPS) from central Chile. BPS were determined as the epidemiologic unit, collecting, for every sampled PS, cloacal or rectal swabs, which were then analyzed by bacterial culture and confirmed by conventional PCR. A positivity rate of 4.17% was estimated among BPS for *S. enterica* in the Metropolitana region, while no positive samples were found in the Valparaíso region. For STEC, a positivity rate among BPS of 11.76% in the Metropolitana region and 18.52% in the Valparaíso region was estimated. The study also identifies different antimicrobial phenotypical resistance profiles in both *S. enterica* and STEC, including multiresistant strains, considered critically important under the One Health approach. The presence of ruminants inside BPS was identified as a factor that raises the risk of positivity for STEC and *S. enterica*/STEC. The study highlights the need for improved biosecurity measures and education regarding zoonotic agents in BPS. The findings of this study are of high importance, providing evidence to policymakers and stakeholders to develop strategies to reduce the risk of transmission of zoonotic agents from BPS to humans and other animals. BPS managed exclusively by women were shown to be at a greater risk for *S. entarica*/STEC positivity compared to men of the family, emphasizing the need to implement and deliver training to women to reduce the current consequences of the gender gap and its potential impact on this animal and human neglected population in Chile.

**Abstract:**

Backyard production systems (BPS) are distributed worldwide, rearing animals recognized as reservoirs of *Salmonella enterica* and Shiga toxin-producing *Escherichia coli* (STEC), both zoonotic pathogens. The aim of this study was to characterize isolates of both pathogens obtained from animals raised in BPS from two central Chile regions. The presence of pathogens was determined by bacterial culture and confirmatory PCR for each sampled BPS, calculating positivity rates. Multivariate logistic regression was used to determine risk factors. Additionally, phenotypic antimicrobial resistance was determined. A positivity rate of 2.88% for *S. enterica* and 14.39% for STEC was determined for the complete study region (Valparaíso and Metropolitana regions). Risk factor analysis suggests that the presence of ruminants (OR = 1.03; 95% CI = 1.002–1.075) increases the risk of STEC-positive BPS, and the presence of ruminants (OR = 1.05; 95% CI = 1.002–1.075) and the animal handlers being exclusively women (OR = 3.54; 95% CI = 1.029–12.193) increase the risk for *S. enterica*/STEC positivity. Eighty percent of *S. enterica* isolates were multidrug resistant, and all STEC were resistant to Cephalexin. This study evidences the circulation of multidrug-resistant zoonotic bacterial strains in animals kept in BPS and the presence of factors that modify the risk of BPS positivity for both pathogens.

## 1. Introduction

*Salmonella enterica* and Shiga toxin-producing *Escherichia coli* (STEC) are zoonotic, Gram-negative bacilli belonging to the *Enterobacteriaceae* family [1,2,3,4]. Transmission for both pathogens occur via the oral–fecal route, by direct or indirect contact with infected animals, fomites, or contaminated foods [2,5,6,7,8,9,10]. They can be present in various reservoirs, some of which are commonly found in rural households, such as poultry, pigs, and ruminants [11,12,13,14,15,16,17,18]. Both *Enterobacteriaceae* can cause severe, life-threatening clinical disease in humans, particularly in the at-risk population, which includes children under 5 and 10 years old, pregnant women, elders, and immunocompromised patients [2,19,20,21,22,23,24].

In developed countries, *S. enterica* and STEC are the second and third leading causes of foodborne disease (FBD) outbreaks, respectively [8,9,25], while in Chile, they are the first and fifth causes, respectively [26,27].

Backyard production systems (BPS) correspond to small-scale agricultural systems with extensive or semi-intensive characteristics, being majorly present in rural and low-income areas [28,29,30]. Most of them are mixed systems, which perform forestry, agriculture and/or livestock farming, but commonly none of these are the main activity or source of income [16,29,31]. Animals raised in BPS (commonly poultry and swine) tend to have low acquisition and maintenance costs, are of small size, and have short productive cycles, which facilitates their availability for domestic consumption, sale, or exchange when needed [29,32,33,34]. In Chile, BPS are mainly managed by women and about 60% of owners are over 55 years old, which increases the likelihood of suffering comorbidities that lead to immunosuppression [30,35,36]. Therefore, most owners are considered in the at-risk population. Additionally, animal and human populations that reside in BPS are frequently neglected.

BPS commonly have poor or no veterinary attention [31], which implies a lack of adequate diagnosis and treatment of diseased animals. Moreover, an absence of production and sanitary protocols, and knowledge regarding zoonotic agents present in animals [16,33,37], allows pathogen maintenance and transmission in these systems. Infections with these pathogens in animals are often asymptomatic and their shedding is intermittent, which facilitates their persistence and spread to the environment [14,24,38]. In addition, humans and animals from different species, ages, health status, and even BPS interact in these landscapes [16,22,33].

Alegria-Moran et al. [16] described a prevalence of *S. enterica* in BPS of 8.3% in the Metropolitana region and of 6.6% in the Valparaíso region in Chile. Additionally, they reported different factors that increase the risk of *S. enterica* positivity in BPS, including breeding different bird species (odds ratio (OR) = 1.04; 95% CI: 1.01–1.07), carrying out mixed production activities (OR = 5.35; 95% CI: 1.2–27.6), and obtaining replacement animals from external sources (OR = 5.19, 95% CI: 1.4–20.5). On the other hand, STEC prevalence ranging between 0 to 72% has been reported in cattle in different countries [13]. In central Chile, positivity rates of 17% in cattle and 1% in slaughtered pigs were reported for STEC [39].

It is known that the emergence and dissemination of antimicrobial resistance (AMR) in bacteria is considered one of the major threats to public health, affecting human, animal, and environmental health, and, therefore, being identified as one of the major issues for One Health [40,41]. In Chile, information concerning antimicrobial (AM) usage in BPS is scarce. Pavez-Muñoz et al. [31] found, for the first time, AMR in STEC strains obtained from BPS, while also evaluating factors associated with AM usage in BPS from the Metropolitana region, which included recognition of sick animals, presence of neighboring poultry and/or swine BPS, visits of veterinary officials, and close contact between animal species present within BPS. This study also found that AMs were commonly administered without prescription or control overdose and frequency, increasing the probability of generating AMR and residues in derived products [42]. Prior studies from central Chile reported 30 *S. enterica* strains isolated from backyard animals, with 11 of them showing either single drug resistance (SDR) or multidrug resistance (MDR) [43].

Considering the absence of biosecurity measures, the highly variable sanitary conditions, the lack of specific knowledge regarding zoonotic pathogens, the emergence of AMR, and the previously stated background, this is also in frame with the One Health concept, which, among other issues, concerns antimicrobial resistance. The aim of this study was to epidemiologically characterize the *S. enterica* and STEC isolates obtained from BPS in the Valparaiso and Metropolitana regions of Chile.

## 2. Materials and Methods

A cross-sectional study was carried out comprising BPS located in the Metropolitana and Valparaíso regions, recording data concerning sample processing (results of microbiological characterization), epidemiological survey responses (epidemiological characterization), and georeference for each sampled BPS. This also included all data acquired on isolated and confirmed strains of *S. enterica* and STEC (with a total of 5 and 37 isolates, respectively).

In Chile, BPS total over 150 thousand producers; in terms of animal population, it is estimated that they contain over 3.7 million poultry species and 400 thousand pigs, most of them concentrated in the studied regions [44].

### 2.1. Sample Size Calculation

A stratified random sampling with proportional allocation was carried out in the above-mentioned regions, stratifying BPS by provinces. The sample size was calculated using the following Equation (1) [45]:(1)$$n=\frac{Z_{\alpha }^{2}pq}{L^{2}}$$
where n represents the sample size; Zα is the required value for confidence = 1 − α, with α corresponding to the confidence level; Zα is the percentile of a standard normal distribution (1 − α/2); p is the expected prevalence of the pathogen; q is (1 − p); and L is the precision of the estimate or margin of error. Assuming a lack of knowledge about the prevalence of STEC in BPS in central Chile, the sample size was calculated assuming a prevalence of 50%, ensuring the highest possible minimum sample size [45]. A confidence level of 95% and a precision of 5% were also set.

Sampled BPS were selected considering the information obtained from the agricultural census conducted by Instituto Nacional de Estadística [46] regarding BPS distribution. A sample size of 84 and 73 BPS was determined for the Metropolitana and Valparaíso regions, respectively, adding up to a total of 157 BPS.

Additionally, the number of samples to be collected within each BPS was calculated following Equation (2) [45]:(2)$$n=\left(1-\alpha^{\frac{1}{D}}\right)\left(N-\frac{\left(D-1\right)}{2}\right)$$
where n is the sample size; N is the population size; D is the estimated minimum number of diseased animals in the group; and α = 1 − the confidence level. Considering the detection of at least 30% of positive animals and that N equals the number of animals by which BPS are defined, a minimum sample size of 8 animals per BPS was calculated.

### 2.2. Sample Collection and Microbiological Analysis

Samples were collected directly from the cloaca or rectum of sampled animals using sterile swabs with Cary-Blair transport medium (Copan^®^, Brescia, Italy), which were then labeled with the animal species and assigned a code to each BPS. In cases where it was needed, environmental feces samples were collected to achieve the required intra-BPS sample size. These were correspondingly labeled to differentiate them.

Samples were transported and stored at 4 °C at the Centralized Laboratory for Veterinary Research (LaCIV) of the Faculty of Veterinary and Livestock Sciences of the University of Chile until further processing. Sample processing protocols, microbiological analyses, and confirmatory PCR were carried out following protocols previously described [16,31,37,47,48,49,50,51].

### 2.3. Determination of S. enterica and/or STEC Positivity Rate

The positivity rate for *S. enterica*, STEC, and *S. enterica*/STEC (including all BPS that were positive for either of these pathogens) was determined, considering each BPS as an epidemiological unit. In this manner, positivity rates were calculated at both regional and provincial levels using the following Equation (3) [45,52]:(3)$$P=\frac{Positive\;cases}{Population\;at\;risk}$$
where positive cases correspond to the total number of agent-positive BPS (be it for only one agent or both, depending on the case) and the population at risk encompasses the total number of BPS sampled in a given geopolitical unit (region or province).

Additionally, choropleth maps representing BPS positivity rates at province level were constructed for each pathogen, applying a graded color scale based on calculated ranges for BPS positivity rates. Choropleth maps were generated using QGis [53].

### 2.4. Determination of Risk Factors for S. enterica and/or STEC Positivity

A previously validated questionnaire was applied to each BPS [16] to characterize animal management, biosecurity measures applied, and socio-demographic variables affecting each BPS. All BPS managers who agreed to participate did so after signing an informed consent form, complying with the protocols of bioethics and responsibility in scientific research and biosafety.

Three multivariable logistic regression models were constructed in order to evaluate the relationship between potential explanatory variables and BPS positivity for either *S. enterica*, STEC, or both of them (*S. enterica*/STEC model) [45]. This last model was included because, as previously stated, they share common epidemiological characteristics.

In these models, the response variable (Y) is dichotomous, because it can only take two values, where Y = 0 and Y = 1 represent the absence and presence of one or both pathogens in each BPS, respectively.

All variables were subjected to a simple logistic regression, selecting those with a liberal *p*-value of equal to or less than 0.15 for each model. All those that met this criterion were further analyzed using Spearman’s correlation (quantitative variables) and Fisher’s exact test (qualitative variables) to check for collinearity and association between variables, allowing for correction of potential confounding factors.

Subsequently, multivariable models were built using a stepwise backward elimination procedure, removing from the model those variables that, when taken out of the models, did not cause a significant change in its likelihood. This was evaluated using a likelihood ratio test [54], eliminating variables that gave *p*-values higher than 0.05 after being removed from the model. Other criteria used to evaluate the elimination of a variable were associated with the variation caused by the removal of a variable over the coefficients of the rest of the variables, retaining variables that, after being eliminated, caused a change in other variables’ regression coefficients of 20% or more. The convergence of the models was set to a value of epsilon (ε) = $e^{-16}$ to guarantee an adequate level of stringency for the models performed. The goodness of fit of the model to the data was assessed using the Hosmer–Lemeshow test [54,55].

All analyses were performed using R statistical software version 4.2.2 [56] and RStudio version 2022.12.0+353 [57], using the “nlme” [58], “lme4” [59], “car” [60], “ggplot2” [61], and “ResourceSelection” [62] packages.

### 2.5. Determination of Antimicrobial Resistance in S. enterica and STEC Isolates

All *S. enterica* and STEC isolates for which recovery was achieved were evaluated by the minimum inhibitory concentration (MIC) method to determine antimicrobial resistance profiles. In order to calculate MIC, an automated VITEK^®^ 2 system (bioMérieux, Marcy-l’Etoile, France), calibrated with reference strains was used according to manufacturer’s instructions, employing AST-GN98 cards for the evaluation. The antimicrobial panel evaluated consisted of drugs commonly used in veterinary and human medicine, including: aminoglycosides (amikacin and gentamicin); β-lactams (amoxicillin-clavulanic acid, ampicillin, cephalexin, cefovecin, cefpodoxime, ceftazidime, ceftiofur, and imipenem); folate synthesis inhibitors (trimethoprim-sulfamethoxazole); nitrofurans (nitrofurantoin); phenicols (chloramphenicol); quinolones (ciprofloxacin, enrofloxacin, and marbofloxacin); tetracyclines (doxycycline); and also determines the presence or absence of the extended spectrum β-lactamase (ESBL) phenotype of the pathogen under study, using a combination of cefepime, cefotaxime, and ceftazidime alone and in combination with clavulanic acid.

Clinical cut-off values were applied according to the Clinical Laboratory Standards Institute [63] and the European Committee on Antimicrobial Susceptibility Testing [64], considering strains with intermediate susceptibility as resistant. Strains resistant to three or more antimicrobial classes were considered multidrug resistant (MDR) [65].

## 3. Results

### 3.1. S. enterica and STEC Positivity Rate in BPS from Valparaíso and Metropolitana Regions

A total of 139 (89%; 139/157) BPS were sampled, achieving the required number of sampled BPS in the Metropolitana region but only 74% (54/73) of the required samples for the Valparaíso region. This was due to health and movement restrictions associated with the COVID-19 pandemic, managing to collect 44.44% (8/18) of the samples required from the Petorca province and none (0/11) from the Quillota province (Table 1).

The geographical area of study (Metropolitana and Valparaíso regions) showed a positivity rate of 2.88% (4/139) at the BPS level for *S. enterica*. This pathogen was found only in the Metropolitana region, with a positivity rate of 4.71% (4/85) at the regional level. In this region, cases were majorly concentrated in the provinces of Melipilla, Cordillera, and Maipo. Animal species associated with *S. enterica*-positive samples included chickens and a goose. The positivity rate of STEC at the BPS level determined for both regions was 14.39% (20/139). The Metropolitana region presented a positivity rate of 11.76% (10/85), with STEC-positive BPS found in four of the six provinces contained within this region, corresponding to: Melipilla, Cordillera, Maipo, and Chacabuco. On the other hand, the Valparaíso region presented a positivity rate of 18.52% (10/54), with positive BPS found in three of the six provinces studied, corresponding to: San Felipe, San Antonio, and Petorca. STEC-positive animal species included pigs, ducks, chickens, geese, cows, sheep, and goats. When evaluating *Enterobacteriaceae*, the positivity rate for the studied geographical area was 17.27% (24/139). In the Metropolitan region, a positivity rate of 16.47% (14/85) was found, determining the presence of the pathogens in the provinces of Melipilla, Cordillera, Maipo, and Chacabuco. Positivity rate values for *Enterobacteriaceae* in the Valparaíso region were the same as for STEC, since none of the sampled BPS in this region were positive for *S. enterica* (Figure 1; Table 1).

### 3.2. Risk Factor Analysis for S. enterica, STEC, and S. enterica/STEC in BPS from Valparaíso and Metropolitana Regions

The multivariate logistic regression model selected for *S. enterica* is shown in Table 2; the results of the univariable logistic regression analysis can be observed in detail in Appendix A. In this model, only the contact between BPS animals and wild birds was determined as a factor that decreases the risk of positivity to *S. enterica* (odds ratio (OR) = 0.06; 95% confidence interval (95% CI) = 0.00–0.064; *p* = 0.02).

The multivariate logistic regression model selected for STEC is shown in Table 3. In this model, only the presence of ruminants inside the BPS (OR = 1.03; 95% CI = 1.00–1.07; *p* = 0.036) was identified as a factor that raises the risk of positivity to STEC.

The model of multivariate logistic regression selected for *S. enterica*/STEC is shown in Table 4. In this model, two variables were identified as factors that alter the risk for *S. enterica*/STEC positivity in BPS, both increasing it. These variables correspond to the presence of ruminants inside BPS (OR = 1.05; 95% CI = 1.02–1.09; *p* = 0.004) and animal handling being exclusively executed by women (OR = 3.54; 95% CI = 1.03–12.19; *p* = 0.045).

### 3.3. Antimicrobial Resistance Profiles of S. enterica and STEC Isolates

A total of 5/5 *S. enterica* and 14/37 STEC strains were recovered, and minimum inhibitory concentration (MIC) analysis was performed on all of them (Appendix A). *S. enterica* isolates came from two BPS located in Melipilla, one BPS located in Maipo and one BPS located in Cordillera; all provinces belonging to the Metropolitana region. On the other hand, STEC isolates came from three BPS located in Melipilla, two BPS located in Chacabuco and one BPS located in Cordillera. The antimicrobial resistance profiles obtained from MIC analysis can be seen in Table 5.

Five antimicrobial resistance profiles were identified in *S. enterica* isolates. Each of them with a frequency of 20% (1/5). Four of these profiles were identified as MDR, presenting resistance to Betalactams, Tetracyclines, Nitrofurans, and Amphenicols, and one was found to be fully antimicrobial-sensitive. In the case of STEC isolates, two antimicrobial resistance profiles were identified. One had a frequency of 85.7% (12/14) and the other profile showed a frequency of 14.3% (2/14). None of the STEC isolates were found to be MDR.

## 4. Discussion

*Salmonella enterica* and STEC are zoonotic pathogens that can cause potentially severe or lethal disease in at-risk populations. Notably, BPS are important drivers of disease transmission and maintenance for both pathogens. Occasionally, both animal handlers and family members present in these systems may be considered at-risk populations, due to their age range, reproductive status, or presence of comorbidities leading to immunosuppression. These family groups may be exposed to these pathogens due to direct and indirect contact with their animals, feces, and the products they obtain from them.

Poultry is the most frequently reported reservoir for *S. enterica* [15,16,22]. This is consistent with the results obtained in this study, since positivity for *S. enterica* was only observed in poultry. On the other hand, STEC has been described in several reservoir species which could be present in BPS, with ruminants acting as the main reservoirs [12,13,14,66]. In the same manner, this study identified ruminants as the main reservoirs, followed by poultry and pigs.

For the whole area under study, *S. enterica* positivity rate at BPS level was 2.88%, a result similar to the 2.9% reported in San Lorenzo, Paraguay in productions keeping poultry of different stages [67], a situation comparable to BPS in Chile keeping animals of various origins, breeds, and stages. In contrast, this value is higher than that of Entre Rios, Argentina [68], and lower than other reports made around the world, from West Bengal, India [69]; South Australia [70]; northwestern Nigeria [71]; central Ecuador [72]; and Tien Giang, Vietnam [73]. This implies that the health status of *S. enterica* in Chile is good when compared to most countries, which could be explained by variables not analyzed in this study, related to climatic conditions, interaction networks and/or host–pathogen interactions.

In Chile, previously reported data for *S. enterica* positivity rate in the Metropolitana region was 8.3%, while 6.6% was recorded in the Valparaíso region [16]. In contrast, the regional positivity rates determined in this study were lower than those reported previously for both studied regions. At the province level, in the Metropolitan region, the province of Melipilla presented a lower positivity rate than previously reported [16]. Meanwhile, the provinces of Maipo and Cordillera presented values of 6.25% and 20%, respectively; findings not previously described. Contrary to this, the province of Chacabuco in the Metropolitana region, and the provinces of San Antonio (5% and 20%) and San Felipe de Aconcagua (10%) in the Valparaíso region, were negative for *S. enterica* in this study, when previous studies have detected circulation for this pathogen in them [16,74]. It is important to mention that, since a cross-sectional study was carried out, positivity rates values could have been underestimated at both the regional and province level. Therefore, given the low positivity rate reported for *S. enterica* in BPS, it is suggested that future studies reduce the study area in order to increase the number of samples per province or to perform longitudinal epidemiological studies.

In Chile, there are reports of STEC in slaughterhouses, zoos, and livestock farms [39,75,76], but information on BPS animals is scarce. A higher positivity rate for *S. enterica* was expected, considering that poultry were the most frequently raised species in these systems; nevertheless, a STEC positivity rate five times higher than that of *S. enterica* was found, highlighting the importance of investing in more research on this pathogen and its circulation in BPS animals. This finding emphasizes the need to educate people about the latent risks of zoonoses associated with BPS animals, given that clinical cases of STEC in the United States and Chile were associated with contact with infected animals [23,77].

Regarding risk factors, this study suggests that potential contact between BPS animals and wild birds is a factor that reduces the risk of *S. enterica* positivity, although biosecurity measures for commercial farms indicate the opposite [28]. Previous studies in wild bird populations from Chile have found a low prevalence of *S. enterica* [78], although their significance for *Enterobacteriaceae* epidemiology has not been previously addressed in this country. This result could be an indirect indicator of the available surface to the animals for their movement, where they would potentially have contact with wild birds. A larger surface area for movement could imply a greater dispersal of feces, decreasing the probability of contact of a susceptible animal with feces from an infected animal; thus, reducing the infection pressure of the pathogen. Further studies should address the role that wildlife animals play in *Enterobacteriaceae* transmission in BPS. On the other hand, the rearing of ruminant species in BPS was identified as a factor that increases the risk of STEC positivity, an expected result given the high prevalence of STEC described in ruminants [79]. This highlights the importance of identifying, characterizing, and monitoring this pathogen in BPS animal species.

In the *S. enterica*/STEC model, the presence of ruminant species in BPS and the animal handler being a woman increased the risk of positivity. This last point is relevant because the majority of BPS managers are women [37,42], and a large percentage are pensioners (over 60 years old) [31]. In Chile, it is common to find women dedicated mostly to domestic labor and caring for people, especially in rural areas [80].

The dedication of time to caring for others and to domestic chores leads to a reduction in the time spent handling animals. Moreover, it is estimated that the average number of hours worked by rural women was 13 h/day, both inside and outside the home [81]. Additionally, in the forestry and livestock sector, it has been found that women have lesser access to training [82,83,84]. Therefore, the lack of time, due to a work overload, and training for rural women could be limiting their capabilities to manage BPS when compared to men, leaving time for only basic management, such as feeding and releasing/enclosing animals and resulting in reduced or absent biosecurity and hygiene measures. Further data supports gender inequality in rural areas; i.e., the percentage of illiteracy in rural women (70+ years of age) of Chile was over 35% and labor force participation rate for rural women was of 19%, 48 percentage points less than rural men and 19 percentage points less than urban women [83,84]. Currently, in the forestry and livestock sector, the percentage of employed women corresponds to 24.2% of the total number of workers, almost 52 percentage points below that of rural men [81]. In this sense, the results of the present study show an example of a potential threat to public health generated from gender inequality in a rural context. Based on the above, it is necessary to increase access and to focus training and education for rural women in productive and biosafety areas, expanding the real scope of the targeted government programs.

In this study, five strains of *S. enterica* were isolated, with five different antimicrobial resistance profiles, 80% of them being MDR. Among the groups of antimicrobials to which resistance was reported were Betalactams, Tetracyclines, Nitrofurans, and Phenicols; similar results were reported for different animal systems from central Chile [43], which includes *S. enterica* isolates from BPS, also observed in isolates from industrial pigs and chickens in Chile [85]. Within the Betalactams, resistance was reported against Penicillin, first- and third-generation Cephalosporins, and Carbapenemics, all of which are considered critically important in human medicine based on five prioritization criteria according to the World Health Organization (WHO) [86]. In the case of STEC, two AMR profiles were identified for the 14 isolates; none corresponded to MDR, but all showed resistance to Cephalexin, and 14.3% to Chloramphenicol, similar to prior reports from industrial animals in Chile [47]. The resistances reported for STEC in this study are not part of the WHO category of critically important AM for human medicine [86]. The use of AM in STEC infections in humans is considered contraindicated due to their potential to increase the risk of HUS. Antibiotics can eliminate the beneficial intestinal microbiota that competes with STEC and also cause lysis of the bacterial cell well, leading to an elevated release of preformed Stx toxins. Moreover, using AM can induce phage production and *stx* gene expression, further exacerbating the infection [87]. This is particularly significant in rural settings where non-prescribed use is often sought as an alternative [31].

Multidrug-resistant *S. enterica* and STEC isolates have been linked to human outbreaks worldwide; most of them have an animal [88,89,90,91,92], vegetal [93,94,95], or environmental origin [90], causing potential severe clinical outbreak in humans [96], and are detected frequently in developing and undeveloped countries [97]. Evidence in Chile is scarce, but food matrices and water sources have been reported to be positive to MDR enteric pathogens [5,98,99,100]. In addition, the identification of MDR strains is relevant because of the therapeutic limitations it could generate in severe bacterial infections and the possibility of transmission of resistance genes through mobile genetic elements to previously sensitive bacteria of the same genus or others [101]. The diversity of resistance profiles could be attributed to the multiple sources of replacement animals, food, and water, and the use of antimicrobials without veterinary prescription in animals raised in BPS.

Both BPS human and animal populations are neglected. This highlights the need to reduce the gaps in knowledge of the health status of these people so that information is available to future researchers and authorities; thus, the need to increase resources for the characterization and surveillance of these human and animal populations with a One Health approach is emphasized. Furthermore, the transfer of knowledge to the at-risk population should be addressed, particularly the education of children through schools and local media, as they are the most active participants in the education of the family group.

## 5. Conclusions

This study shows the circulation of *S.* enterica and STEC strains in different animal species kept in BPS in central Chile. Additionally, it shows the circulation of strains with different antimicrobial resistance profiles, detecting fully sensitive isolates and others multiresistant to AM which are considered critically important under the One Health approach, because of their use in human bacterial disease treatment and their effects on animal, environmental, and human health.

Furthermore, the existence of factors that modify the risk for BPS positivity for both pathogens in central Chile was also evidenced, highlighting the need to educate BPS owners, especially women, and their families, in topics regarding biosecurity and antimicrobial stewardship, thus adding to the reduction of gender inequalities in rural communities of Chile and decreasing the probability for the emergence of zoonotic pathogens showing AMR, which could threaten animal and human public health in Chile.

## Figures and Tables

**Figure 1 animals-13-02444-f001:**
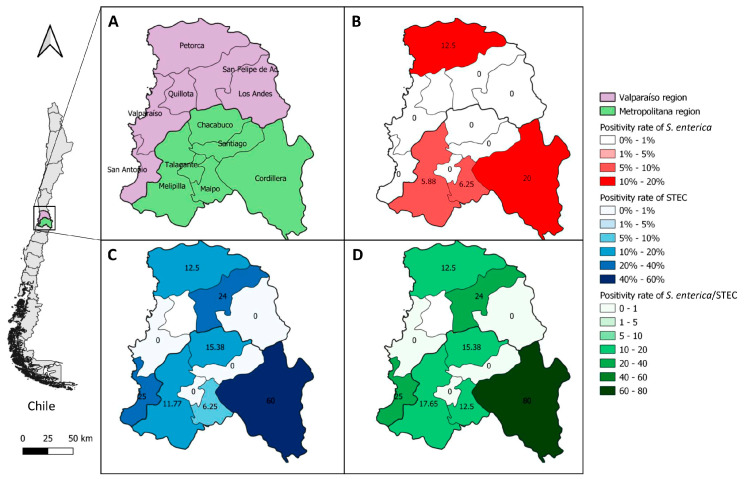
Choropletic maps of *S. enterica*, STEC, and *S. enterica*/STEC positivity rates by province in the Metropolitana and Valparaíso regions of Chile. (**A**) Provinces evaluated in this study and their respective region. (**B**–**D**) shows *S. enterica*, STEC, and *S. enterica*/STEC positivity rates by province, respectively.

**Table 1 animals-13-02444-t001:** Backyard production systems sampled and positivity rate by province and region.

Region	Province	N° of BPS Sampled	N° of *S. enterica*-Positive BPS	Positivity Rate of *S. enterica* (%)	N° of STEC-Positive BPS	Positivity Rate of STEC (%)	N° of *S. enterica*/ STEC-Positive BPS	Positivity Rate of *S. enterica*/ STEC (%)
Metropolitana	Melipilla	34	2	5.88%	4	11.77%	6	17.65%
	Talagante	7	0	0%	0	0%	0	0%
	Cordillera	5	1	20%	3	60%	4	80%
	Maipo	16	1	6.25%	1	6.25%	2	12.50%
	Chacabuco	13	0	0%	2	15.38%	2	15.38%
	Santiago	10	0	0%	0	0%	0	0%
	Subtotal	85	4	4.71%	10	11.76%	14	16.47%
Valparaíso	Los Andes	3	0	0%	0	0%	0	0%
	de Valparaíso	6	0	0%	0	0%	0	0%
	San Felipe	25	0	0%	6	24%	6	24%
	San Antonio	12	0	0%	3	25%	3	25%
	Petorca	8	0	0%	1	12.50%	1	12.50%
	Quillota	0	0	0%	0	0%	0	0%
	Subtotal	54	0	0%	10	18.52%	10	18.52%
	Total	139	4	2.88%	20	14.39%	24	17.27%

**Table 2 animals-13-02444-t002:** Logistic regression model associated with risk factor determination for *S. enterica* positivity in BPS.

Variable	Category	*p*-Value	OR ^1^	95% CI ^2^	
				Lower	Upper
(Intercept)	-	0.352	0.486	0.106	2.221
N° of pets	-	0.160	0.541	0.229	1.277
Animals have contact with wild birds	No	reference			
	Yes	0.019	0.059	0.005	0.636

^1^ Odds ratio; ^2^ confidence interval.

**Table 3 animals-13-02444-t003:** Logistic regression model associated with risk factor determination for Shiga toxin-producing *Escherichia coli* positivity in BPS.

Variable	Category	*p*-Value	OR ^1^	95% CI ^2^	
				Lower	Upper
(Intercept)	-	<0.001	0.027	0.003	0.218
Presence of ruminants	-	0.036	1.038	1.002	1.075
Presence of guinea pigs or rabbits	-	0.132	1.126	0.964	1.316
Animals have contact with wild birds	No	reference			
	Yes	0.251	3.429	0.417	28.179
Receive state assistance or support	No	reference			
	Yes	0.266	1.901	0.612	5.894

^1^ Odds ratio; ^2^ confidence interval.

**Table 4 animals-13-02444-t004:** Logistic regression model associated with risk factor determination for *S. enterica*/STEC positivity in BPS.

Variable	Category	*p*-Value	OR ^1^	95% CI ^2^	
				Lower	Upper
(Intercept)	-	<0.001	0.027	0.003	0.218
Presence of ruminants	-	0.004	1.038	1.002	1.075
Person in charge of the system	Family	reference			
	Man	0.213	2.422	0.603	9.732
	Woman	0.045	3.542	1.029	12.193

^1^ Odds ratio; ^2^ confidence interval.

**Table 5 animals-13-02444-t005:** Antimicrobial resistance profiles identified in *S. enterica* and STEC isolates, according to MIC results.

Pathogen (N° of Isolates)	N° of Isolates	Frequency	Antimicrobial Resistance Profile	N° of Antimicrobial Drugs with Strain Resistance
*S. enterica* (5)	1	20%	CLX, IMP, DXC, NTF *	4
	1	20%	CLX, CVN, CPN	3
	1	20%	AMP, AMX, DXC, NTF, CPN	5
	1	20%	AMP, AMX, CTD, DXC, NTD, CPN	6
	1	20%	-	0
STEC (14)	12	85.7%	CLX	1
	2	14.3%	CLX, CPN	2

* CLX: Cephalexin, IMP: Imipenem, DXC: Doxycycline, NTF: Nitrofurantoin (NTF), CVN: Cefovecin, CPN: Chloramphenicol, AMP: Ampicillin, AMX: Amoxicillin with clavulanic acid, and CTD: Ceftazidime.

## Data Availability

The data presented in this study are available on request from the corresponding author. The data are not publicly available because they are part of an ongoing project not yet published.

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
