# Peer review of "Epidemiological Characterization of Isolates of Salmonella enterica and Shiga Toxin-Producing Escherichia coli from Backyard Production System Animals in the Valparaíso and Metropolitana Regions"

_animals, 2023, doi:10.3390/ani13152444_

Round 1

Reviewer 1 Report

A very sound and competent epidemiological study. The risks posed by zoonotic pathogens by Backyard production in LMICs is poorly understood. Whilst the findings apply on really to Chile/Latin America directly, the work shows there is a background prevalence and risk of foodborne infection (and development of AMR) even in extensive production  which has been shown in Africa and Asia.

Author Response

We greatly appreciate your comments. Please see the attachment for details.

Reviewer 2 Report

Title: Epidemiological Characterization of Isolates of Salmonella enterica and Shiga toxin-producing Escherichia coli from Backyard 3 Production Systems Animals in the Valparaíso and Metropoli tana regions.

General observations: The study provides valuable insights into the prevalence of Salmonella and STEC in backyard production systems in Chile, as well as the phenotypic antibiotic resistance exhibited by a subset of isolates. It is crucial to continue generating additional information from Latin America, considering the limited published data available from low-income and rural areas. However, to enhance the publication, particular attention should be given to improving the methodology description and presenting more comprehensive results. The inclusion of detailed information regarding the studied locations and the exploration of univariate associations can offer a better understanding of the potential factors associated with the prevalence of these pathogens.

Moreover, the discussion section would benefit from incorporating additional evidence that supports the occurrence of zoonotic events in similar settings. By providing further substantiation of zoonotic transmission and related implications, the study can contribute to the broader understanding of public health risks in these particular contexts.

Specific suggestions:

       Title: Shigatoxin-producing, separate Shiga toxin

       Simple summary: It is suggested to clarify the “positivity rate of 4.17% for Salmonella enterica in the Metropolitana region ….”, indicating whether it refers to the percentage within the backyard production systems (BPS) or among BPSs. Additionally, it is unclear from the summary what the unit of analysis is (e.g., environment, soil, water, animal feces, owners’ feces). Moreover, it is recommended to provide further clarification regarding the suggestion of training women, as it is not evident whether any gender associations were found.

       Abstract: The mention of "wild birds decreasing the risk of S. enterica" needs further exploration, possibly through the inclusion of additional analysis to identify if this is the result of confounding factors. Consequently, I will exclude it from the abstract. OR should include CI.

       In Line 160, the reference [31] is provided for the description of methods, but it is noted that the referenced article only covers methods for STEC and not Salmonella. Therefore, a more appropriate reference should be used, or the methods should be better described in this section.

       Line 172, mention the software utilized for generating choropleth plots.

       Line 205, it is recommended to include citations for the statistical analysis packages employed in R.

       Throughout the paper the term "Enterobacteriaceae" used to refer to co-detection of Salmonella and STEC is deemed inappropriate, as it may include other genera and non-pathogenic E. coli. It is advised to revise the terminology accordingly.

       Provide a better description of Metropolitana and Valparaiso, such as demographics and weather and why these two locations were chosen. Furthermore, it is necessary to report the characteristics of BPSs sampled in this study, including the number of animal species reported and the distribution of BPS with respect to having one or more than one animal species. This information could provide more insight into why some differences were observed between them.

       Furthermore, the study should consider including additional demographic factors that could provide a more comprehensive picture of the studied population. It is not mentioned whether the households have access to drinking water, nor are other crucial factors, such as socio-economic level, the number of animals in the household, or the level of education of the participants, addressed. These factors can significantly influence the dynamics of enteric pathogens and antimicrobial resistance within the population and should be considered for a more robust analysis.

       The association between wild bird contact and S. enterica detection should be further investigated as it is suggested that a confounding factor may be involved.

       To provide a comprehensive understanding of the studied population, it is recommended to report univariate analyses in a table format exploring the associations between the detection of Salmonella/STEC and categorical variables, with particular emphasis on variables such as the presence of ruminants, gender handling, wild bird contact, and poultry.  The table should contain odds ratios (OR) with 95% confidence intervals (CI), and p-values calculated using either the chi-square or Fisher's exact test. Additionally, comparing BPS with one animal species versus more than one can provide valuable insights.

       Table 1 shows that 4 BPS were positive for Salmonella and 20 for STEC, but it is requested to specify the number of isolates obtained in each location and from which animal species. To address this, it is recommended to include a supplementary table indicating the source of strains by animal species. Following this, in Line 290, it can be reported as "5/XX strains" and "14/XX strains," while also mentioning how the strains were selected for the resistance test, such as randomly.

       Table 5: The heading "N° of antimicrobial resistance" should be revised to "N° of antimicrobial drugs with strain resistance" to accurately represent the content of the table.

       The discussion section can be enhanced by providing additional evidence of zoonotic events involving enteric pathogens and antimicrobial resistance in similar settings, particularly in low-income and rural areas.

       In Line 416, it is necessary to state that antibiotics are not recommended to treat STEC infections due to the potential worsening of symptoms. It is important to emphasize that the detection or presence of antibiotic-resistant STEC strains becomes relevant in the context of responsible antibiotic use in animals.

       To provide a comprehensive understanding of the studied population and potential sources of antibiotic resistance genes, it is crucial to investigate if there is significant animal production in the areas under study (i.e., big poultry production). It would be beneficial to gather information on whether families were queried about their use of antibiotics in animals, including the types of antibiotics used. Additionally, it is relevant to explore whether local veterinary centers or animal stores sell antibiotics to families without requiring a prescription. Understanding the availability and usage of antibiotics in the community is particularly important given the emergence of multidrug-resistant (MDR) Salmonella, which poses a significant threat. Collecting more information on litter disposal practices and the quality of served water in the studied population can also contribute to a better understanding of the overall context.

Author Response

We greatly appreciate your comments. Please see the attachment for details

Reviewer 3 Report

This study examined 139 backyard production systems (BPS) in central Chile, identifying 4 strains of S. enterica and 20 strains of Shigatoxin-producing Escherichia coli (STEC) isolates. Risk factor analysis indicated that contact between BPS animals and wild birds reduces the risk of S. enterica-positive BPS, while the presence of ruminants increases the risk of STEC-positive BPS. Additionally, the presence of ruminants and exclusive female animal handlers increases the risk of Enterobacteriaceae positivity. Minimum inhibitory concentration (MIC) analysis revealed that 80% of S. enterica isolates were multidrug resistant, and all STEC isolates displayed resistance to Cephalexin.

While this study holds significance, certain issues need to be addressed.

1. Line 160: The manuscript mentions that sample processing protocols and microbiological analyses are described in reference [31]. However, as reference [31] primarily focuses on STEC, it is crucial to provide details regarding the protocol used to differentiate and identify the isolated bacteria as either STEC or S. enterica. This information is essential for the proper characterization of the isolates.

2. The central focus of this study revolves around the isolation of STEC and S. enterica strains from central Chile. It is imperative to present the results of culture-specific and biochemical traits or genome-wide characterization analyses. This would provide insights into the methods employed to determine the classification of these isolates as STEC or S. enterica.

3. It is important to clarify the total number of S. enterica and STEC strains that were isolated. According to Line 121, a total of 5 S. enterica isolates and 37 STEC isolates were obtained. Line 240 states a positivity rate of 2.88% (4/139) for S. enterica, and Line 244 reports a positivity rate of 14.39% (20/139) for STEC in both regions. These figures should be explicitly mentioned for clarity.

4. The manuscript states that minimum inhibitory concentration (MIC) analysis was performed on 5 S. enterica and 14 STEC recovered isolates. However, it is not specified whether all the STEC isolates were subjected to MIC analysis. It is essential to address whether the remaining STEC isolates underwent similar analysis and, if not, provide a rationale for the selection of specific isolates for MIC testing.

Minor editing of English language required.

Author Response

(The authors gave the same response as above.)

Reviewer 4 Report

This manuscript covers the shedding of Salmonella and STEC in backyard production systems in two areas of Chile. It contains some potentially useful data. This has the potential to be an interesting manuscript, however it is lacking detail in places, and the discussion became relatively repetitive and long.

I have detailed specific comments below

Line 42- this is the first and only time that PCR is mentioned. This needs detailing in the methods section

Line 57- Gram needs capitalising

Line 78- considered to the in the at risk population …. (reword)

Line 83- instead of ease, perhaps allow or encourage may work better?

Line 97-105- this is very difficult to follow. Please reword

Line 108- Salmonella enterica is written in different ways throughout, please be consistent

Line 112- frame with the One Health concept …. (reword)

Line 113- concerns antimicrobial resistance …. (reword)

Line 114- was to epidemiologically and microbiologically characterise …. (reword)

Line 156- indicating that they correspond to environmental …. (reword)

Methodology- please include manufacturers of reagents where possible please

Line 160- this needs much more detail here as the manuscript is difficult to decipher without it

Line 180- please link these two sentences or they do not make sense otherwise

Line 186-188- again, please link these two sentences or they do not make sense otherwise

Line 200- rest of the variables. … (reword)

Line 201- coefficients of 20% or more …(Reword)

Line 208- how were these chosen?

Line 211- which reference strains were used?

Line 212-219- please include concentrations of antibiotics

Line 223- taking the samples as resistant if classes as intermediate is not conservative?!

Line 290-291- how were these isolates chosen?

Table 5- typo in frequency

Line 3331- while 6.6% has been recovered din the Valparaiso region and 2.6% in the ,… (reword)

Line 347- higher than that reported …(reword)

Line 351- lower than that reported in … (reword)

The discussion is over long and repetitive in many places, please cut this down. A table may help to include the prevalence values as that takes up a lot of space

Line 372- delete animals

Line 433- what is AM? This is the first abbreviation of this

Line 436-440- this is unclear-please reword

A few comments have been made above

Author Response

(The authors gave the same response as above.)

Round 2

Reviewer 3 Report

Thank you for addressing all of my concerns. I don't have any suggestions or comments at this time.

Reviewer 4 Report

I wish to thank the authors for addressing my comments and wish them the best of luck with their future research